# Membrane fluidity control by the *Magnaporthe oryzae* acyl-CoA binding protein sets the thermal range for host rice cell colonization

**Michael Richter**[1], **Lauren M. Segal**[1], **Raquel O. Rocha**[1¤], **Nisha Rokaya**[1], **Aline R. de Queiroz**[2], **Wayne R. Riekhof**[3], **Rebecca L. Roston**[2], **Richard A. Wilson**[1] *

1 Department of Plant Pathology, University of Nebraska-Lincoln, Lincoln, Nebraska, United States of America, 2 Center for Plant Science Innovation, Department of Biochemistry, University of Nebraska-Lincoln, Lincoln, Nebraska, United States of America, 3 School of Biological Sciences, University of Nebraska-Lincoln, Lincoln, Nebraska, United States of America

¤ Current address: Department of Plant Pathology and Ecology, The Connecticut Agricultural Experiment Station, New Haven, Connecticut, United States of America
* rwilson10@unl.edu

**Data Availability Statement:** The strains generated during the course of this study are available from the corresponding author upon request and with a

## Abstract

Following leaf cuticle penetration by specialized appressorial cells, the devastating blast fungus *Magnaporthe oryzae* grows as invasive hyphae (IH) in living rice cells. IH are separated from host cytoplasm by plant-derived membranes forming an apoplastic compartment and a punctate biotrophic interfacial complex (BIC) that mediate the molecular host-pathogen interaction. What molecular and cellular processes determine the temperature range for this biotrophic growth stage is an unanswered question pertinent to a broader understanding of how phytopathogens may cope with environmental stresses arising under climate change. Here, we shed light on thermal adaptation in *M. oryzae* by disrupting the *ACB1* gene encoding the single acyl-CoA-binding protein, an intracellular transporter of long-chain acyl-CoA esters. Loss of *ACB1* affected fatty acid desaturation levels and abolished pathogenicity at optimal (26˚C) and low (22˚C) but not elevated (29˚C) infection temperatures (the latter following post-penetration shifts from 26˚C). Relative to wild type, the Δ*acb1* mutant strain exhibited poor vegetative growth and impaired membrane trafficking at 22˚C and 26˚C, but not at 29˚C. *In planta*, Δ*acb1* biotrophic growth was inhibited at 26˚C–which was accompanied by a multi-BIC phenotype—but not at 29˚C, where BIC formation was normal. Underpinning the Δ*acb1* phenotype was impaired membrane fluidity at 22˚C and 26˚C but not at elevated temperatures, indicating Acb1 suppresses membrane rigidity at optimal- and suboptimal- but not supraoptimal temperatures. Deducing a temperature-dependent role for Acb1 in maintaining membrane fluidity homeostasis reveals how the thermal range for rice blast disease is both mechanistically determined and wider than hitherto appreciated.

suitable APHIS permit. All raw data supporting the findings of this study are available in S4 Table. All experiments were comprised of at least three biological replicates, based on precedence in the field. Figure legends specify the number of biological and technical replicates, and the statistical tests used in each experiment.

**Funding:** This work was supported by National Science Foundation funding (IOS- 2106153 to R.A. W.) R. R. was partially supported by the Nebraska Agricultural Experiment Station with funding from the Hatch Multistate capacity funding program (Accession Number NEB-30-131) from the USDA National Institute of Food and Agriculture. The funders had no role in study design, data collection and analysis, decision to publish, or preparation of the manuscript.

**Competing interests:** The authors have declared that no competing interests exist.

## Author summary

A major challenge to global food security lies in anticipating phytopathogen spread due to climate change. Relatedly, how temperature affects pathogen-host plant interactions at the molecular level is poorly understood. Here, using molecular, cellular and biochemical approaches, we investigated thermal adaptation by the devastating rice blast fungus *Magnaporthe oryzae*. We show that mediation of fungal membrane fluidity by the intracellular lipid-intermediate transporter Acb1 is critical for both axenic growth and host plant cell-associated invasive hyphal growth at optimal and low environmental temperatures. However, at elevated temperatures, Acb1 is dispensable. By uncovering Acb1 as a temperature-dependent determinant of plant cell colonization, we reveal the molecular strategies by which a fungal pathogen adapts to environmental temperatures during host infection.

## Introduction

To safeguard agriculture, understanding the effects of climate change on pathogen migration and the emergence of new plant pathogens is critical. One aspect of this problem is the need to know how plant-pathogen interactions are affected by environmental factors including temperature extremes [1–4] and thus to define what, at the molecular and cellular level, determines the range of climatic conditions under which pathogens can infect hosts. Here, in the devastating rice blast fungus *Magnaporthe oryzae* (syn. *Pyricularia oryzae*), we shed light on these questions by providing a mechanistic account of pathogen thermal adaptation during host plant infection.

*M. oryzae* is an ascomycete fungus that exhibits a hemibiotrophic infection cycle such that it colonizes living host rice cells as a biotroph for the first few days of infection before symptoms develop and the fungus enters its necrotrophic growth phase [5–8]. Infection begins when a spore germinates on the leaf surface and forms a specialized infection cell called an appressorium that penetrates the rice cuticle, allowing invasive hyphae (IH) to elaborate and fill the first underlying epidermal cell before spreading via pit field sites into neighboring, living cells at around 44 hours post inoculation (hpi) [9–13]. During this biotrophic growth stage, IH are surrounded by the plant-derived extra-invasive hyphal membrane (EIHM), forming a compartment into which apoplastic effectors like Bas4 are secreted via conventional ER-Golgi vesicular transport. In addition, a single vesicle-rich biotrophic interfacial complex (BIC) forms at the tip of the first invasive hyphal strand before migrating to a subapical position in the first infected rice cell—and at the tips of new IH spreading in neighboring cells— and is comprised of plant lipids [7,14]. The BIC is outside the fungal cell wall and receives cytoplasmic effectors such as Pwl2, which are secreted into the BIC via an unconventional pathway involving a t-SNARE and components of the exocyst complex, before they are translocated into rice cytoplasm in a manner involving host clathrin-mediated endocytosis [7,14,15]. Although BICs comprise of host membranes, loss of the *M. oryzae* exocyst component Sso1 leads to a double-BIC phenotype [14], while changes to *PWL2* codon usage can inflate the BIC due to unregulated Pwl2 synthesis and secretion [15], together indicating that fungal processes contribute to BIC membrane dynamics and structure. Furthermore, the maintenance of EIHM and BIC integrity during biotrophy depends on fungal membrane homeostasis mediated by *M. oryzae* autophagy [16,17].

This study was motivated by a desire to better understand fungal membrane homeostasis during biotrophy. Based on known functions in yeast, we hypothesized that the *M. oryzae*

acyl-CoA binding protein-encoding *ACB1* homologue may be involved. Acb1 transports acyl-CoA esters from fatty acid synthetase to acyl-CoA-requiring processes. In yeast, Acb1 is required for fatty acid elongation and desaturation, membrane assembly and organization, and membrane trafficking [18–22]. While *ACB1* was recently shown to be necessary for virulence in *M. oryzae* [23], its function in fatty acid composition and membrane trafficking was unknown. To test our hypothesis that *M. oryzae ACB1* is involved in membrane homeostasis, we targeted it for disruption and characterization. Our results show that *M. oryzae ACB1* is required for axenic and biotrophic growth—including BIC organization—at sub-optimal (22°C) and optimal (26°C) but not elevated (i.e. supraoptimal) temperatures (29°C) due to its role in membrane fluidity control. In determining a novel temperature-dependent role for *ACB1* in thermal adaptation, we demonstrate how *ACB1* establishes the temperature range for host plant infection, which is wider than previously reported once the host has been successfully penetrated.

## Results

### *M. oryzae ACB1* is required for lipid homeostasis, biotrophic growth and BIC organization

We disrupted the *M. oryzae* locus MGG_06177, containing the single Acb1-encoding homolog, using split-marker-mediated homologous recombination [24]. This replaced the 5' end of the *ACB1* coding sequence with the *ILV1* gene conferring sulphonyl urea resistance. Deletants were confirmed by PCR and three independent mutant strains were tested and shown to have identical phenotypes. One Δ*acb1* deletant was used to receive the *ACB1* gene under its native promoter to generate the Δ*acb1 ACB1* complementation strain. The same Δ*acb1* deletant was separately transformed with the pBV591 vector [25] expressing, under native promoters, *PWL2-mCherry:NLS* and *BAS4-GFP* to produce the apoplastic effector Bas4 fused to GFP and the cytoplasmic effector Pwl2 fused to mCherry and a nuclear localization signal (NLS).

We first sought to determine if the loss of *M. oryzae ACB1*, like in yeast [19], perturbed lipid homeostasis by using gas chromatography-mass spectrometry (GC-MS) to analyze fatty acid methyl esters (FAMEs) extracted from vegetative mycelia grown in liquid minimal media (MM) under our standard conditions of 26°C for 40 h. In Acb1-depleted yeast cells, the total cellular content of fatty acids was reduced compared to WT, but the relative levels of unsaturated fatty acids were increased, except for oleic acid ($C_{18:1}$), which was reduced [18,19,22]. Our results, averaged from three biological replicates per sample (Fig 1A, raw values in S1 Table), generally concur with those reported from yeast [19,22]: *M. oryzae* Δ*acb1*, compared to WT and the Δ*acb1 ACB1* complementation strain, was significantly ($P < 0.05$) reduced for total cellular fatty acid content overall (Fig 1A, *left*), and its relative levels of $C_{18:1}$ were one quarter those of WT (Fig 1A, *right*). However, although the relative levels of linoleic acid ($C_{18:2}$) were similar in both WT and Δ*acb1*, the relative amount of α-linolenic ($C_{18:3}$) was increased 3-fold in Δ*acb1* compared to WT and the complementation strain (Fig 1A, *right*). Levels of stearic acid ($C_{18:0}$) were also increased by 3.7-fold in Δ*acb1* mycelia compared to WT and the complementation strain, while the relative abundances of palmitic acid ($C_{16:0}$) were decreased two-fold in Δ*acb1* relative to WT, the latter indicating changes to the acyl-CoA chain length profile as described in yeast [19]. Thus, the loss of *ACB1* in *M. oryzae* decreased the total cellular content of fatty acids and altered the relative level of most measured fatty acids compared to WT (notably $C_{18:3}$), indicating that, like in yeast, *M. oryzae ACB1* is required for lipid homeostasis.

At the standard 26°C infection temperature, the *M. oryzae* Δ*acb1* mutant strain was non-pathogenic on whole plants compared to WT and the Δ*acb1 ACB1* complementation strain

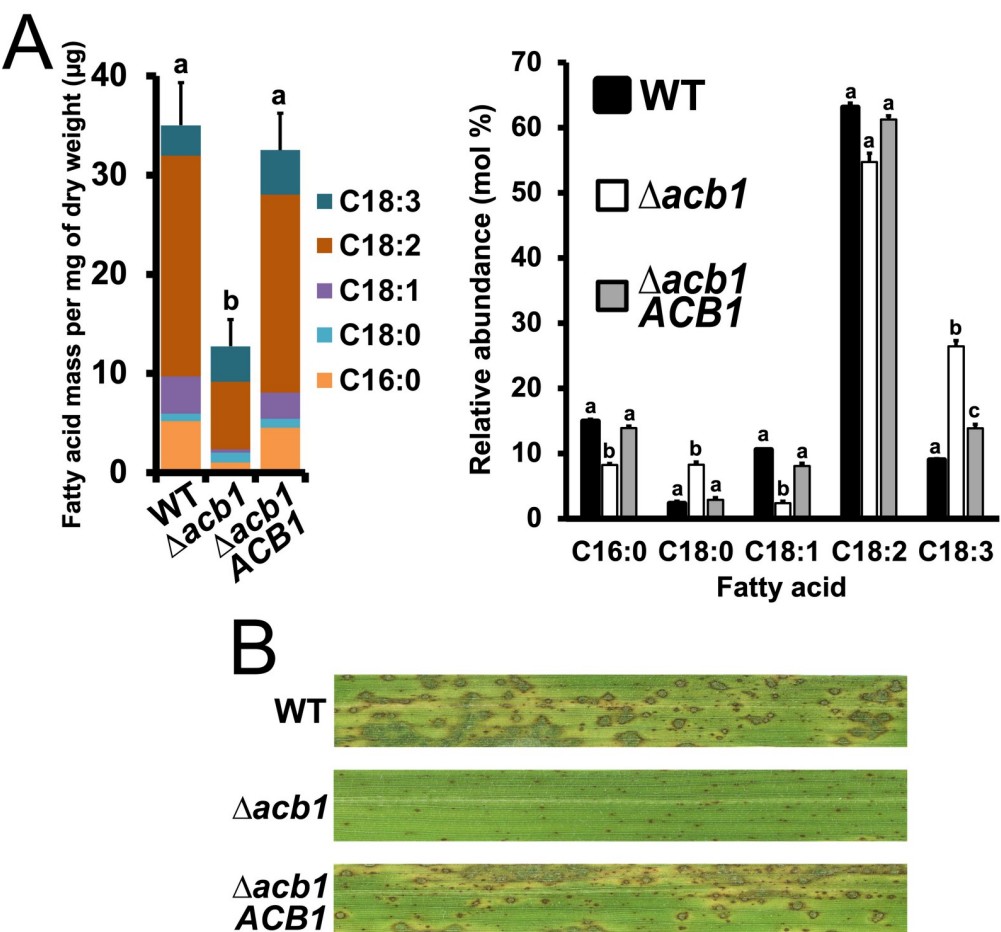

**Fig 1. *M. oryzae ACB1* is required for lipid homeostasis, pathogenicity on rice, and BIC organization. A**. Fatty acid methyl ester (FAME) analysis by GC-MS showing the total (*left*) and relative (*right*) abundances of the five most abundant fatty acids in each strain. Samples were extracted at 40 h post inoculation (hpi) from vegetative mycelia incubated for 24 h in liquid complete media followed by 16 h in liquid minimal media at 26°C. Bars depicts the means of three biological replicates (S1 Table). Error bars are SD. Different lowercase letters indicate statistical differences as determined by one-way ANOVA followed by an unpaired t test with Bonferroni correction, $P < 0.05$. **B**. Pathogenicity assay of the Guy11 wild-type (WT) strain, the *ACB1* gene deletion strain Δ*acb1*, and the Δ*acb1 ACB1* complementation strain. Images are representative of at least 5 leaves from 3 independent biological replicates and were taken of 3-weeks-old CO-39 rice seedlings at 120 hpi. Spores of the indicated strains were inoculated at a rate of $1 \times 10^5$ spores ml$^{-1}$.

(Fig 1B). Consistent with previous work [23], appressorial cell collapse rates calculated using the incipient cytorrhysis assay [12] showed Δ*acb1* appressoria were reduced for turgor compared to WT at this temperature (S1A Fig). Nonetheless, the Δ*acb1* mutant strain formed appressoria that penetrated the host cuticle (see below), while live-cell imaging of detached rice leaf sheaths at 36 hpi (Fig 2A and 2B), using a Δ*acb1* strain producing the apoplastic effector Bas4 fused to GFP and the cytoplasmic effector Pwl2 fused to mCherry:NLS, showed that Δ*acb1* elaborated IH in underlying rice epidermal cells. Furthermore, at 36 hpi (Fig 2A and 2B), the EIHM and apoplastic space of Δ*acb1* were, like those of the isogenic *ACB1*⁺ strain, intact in the first infected rice cell. However, unusual dual-BIC (Fig 2A) or multi-BIC (Fig 2B) structures occurred stochastically and with high frequency in Δ*acb1*-infected rice cells compared to WT-infected rice cells (quantified in Fig 2C). Also, Bas4 was often retained in intracellular compartments in Δ*acb1* IH in the first infected rice cell (Fig 2A and 2B), suggesting

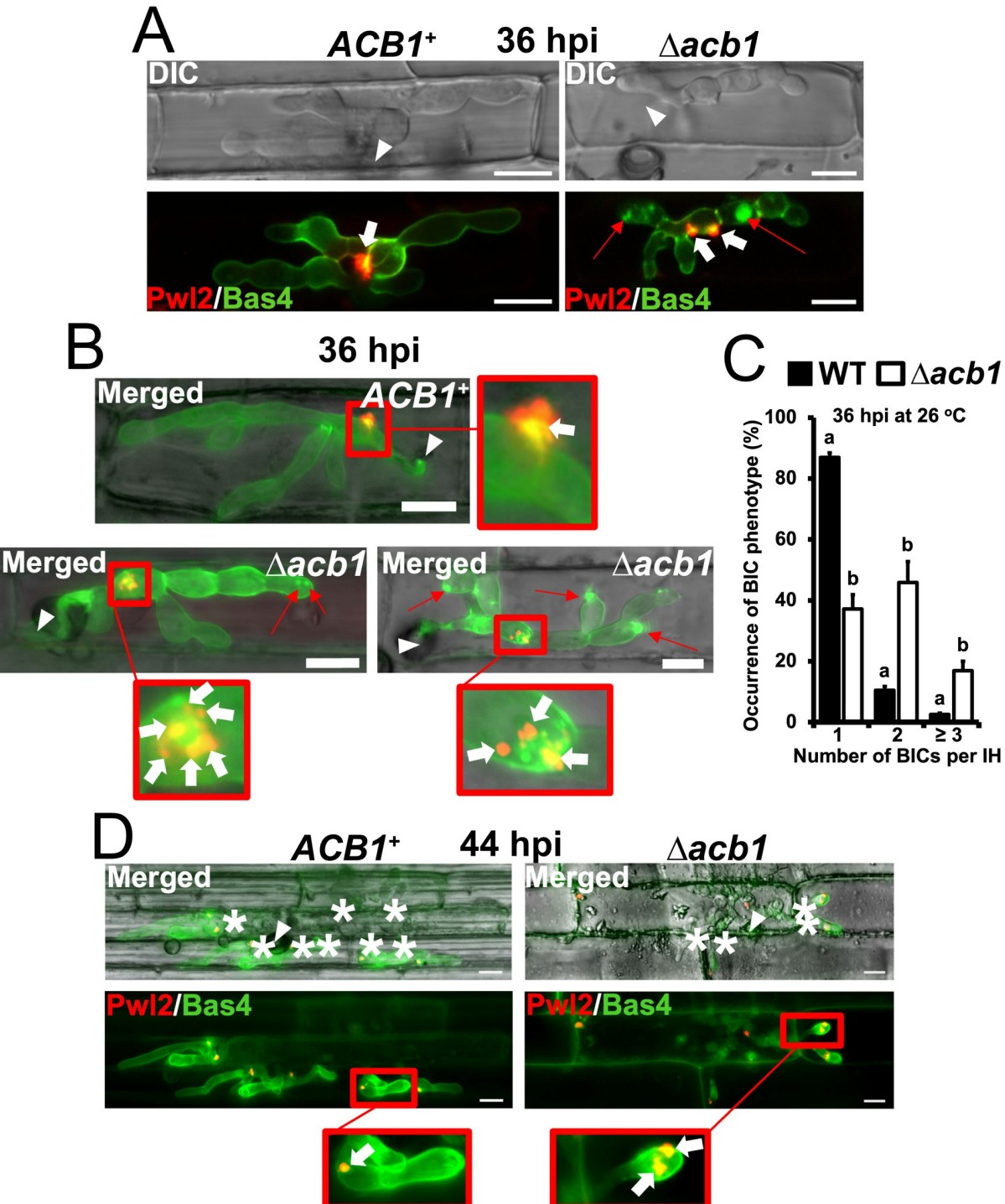

**Fig 2. *ACB1* prevents stochastic loss of BIC integrity. A,B**. Live-cell imaging at 36 hpi of detached rice leaf sheath epidermal cells infected with the indicated strains producing BIC-localized Pwl2-mCherry:NLS and apoplast-localized Bas4-GFP. **C**. Percentage of IH of each strain displaying the indicated number of BICs/ BIC fragments in the first infected rice cell at 36 hpi. Detached rice leaf sheaths were inoculated with $1 \times 10^5$ spores ml$^{-1}$ and incubated at 26°C. Bars depict means ± SD and were calculated from observing 50 infected rice cells per treatment, repeated in triplicate. Different lowercase letters denote statistical differences (one-way ANOVA, Welch's t test, $P < 0.05$). **D**. Live-cell imaging at 44 hpi of detached rice leaf sheath epidermal cells infected with the indicated strains producing BIC-localized Pwl2-mCherry:NLS and apoplast-localized Bas4-GFP. **A,B, D**. Leaf sheaths were inoculated with $3 \times 10^4$ spores ml$^{-1}$ and incubated at 26°C. White arrowheads indicate sites of appressorial penetration, white arrows show the location of the BICs, thin red arrows indicate the retention of ER-Golgi secreted Bas4-GFP in intracellular compartments, and asterisks indicate where IH have moved to neighboring cells; scale bars, 10 μm. Images are representative of 50 infected rice cells per leaf sheath per strain, repeated in triplicate.

membrane vesicle trafficking through the ER-Golgi network was impaired. By 44 hpi, compared to WT, Δ*acb1* IH did not fill the first infected cell and only a few strands of Δ*acb1* IH migrated into neighboring cells (Fig 2D, quantified below). These migrating Δ*acb1* IH often formed dual-BICs at new IH tips, in contrast to WT which only formed single tip-BICs (Fig 2D). Together, our first results showed that *ACB1* is a pathogenicity determinant required for biotrophic growth and BIC organization in host rice cells.

## *M. oryzae ACB1* is required for thermal adaptation and membrane trafficking at (sub)optimal but not supraoptimal temperatures

To better understand how the loss *ACB1* impacted biotrophic growth, BIC integrity and the trafficking of Bas4-containing secretory vesicles, we next hypothesized that the loss of *ACB1* affected thermal adaptation by the fungus. Although not examined in yeast, this hypothesis was formulated based on our preliminary observations of plate growth tests, where we noted that by 10 days post-inoculation, Δ*acb1* not only grew less than WT on complete media (CM) at 26˚C (which, in our hands, is the optimal media and temperature for *M. oryzae* growth and sporulation), but also grew much worse at room temperature (22–24˚C). To test our hypothesis, we grew biological replicates of Δ*acb1* and WT on complete media and formally compared Δ*acb1* and WT vegetative growth at suboptimal (22˚C), optimal (26˚C) and supraoptimal (29˚C—32˚C) temperatures. Fig 3A shows how Δ*acb1* colony diameters were reduced compared to WT at 26˚C and at 22˚C (quantified in S2A Fig). However, Δ*acb1* growth rates were indistinguishable from WT at 29˚C, with both strains growing comparable to WT at 26˚C, and at 30˚C and 32˚C, where both strains grew similarly to each other but reduced compared to growth at 29˚C (Fig 3A) (30˚C results are included here to show how radial growth for both strains peaked at 29˚C, but 30˚C was not used in subsequent analyses). Similar results were obtained on minimal media (MM) (S2B Fig). We also harvested spores from colonies on 10-day-old plates, finding that Δ*acb1* sporulation rates were reduced compared to WT on a per plate basis at 26˚C (as previously reported [23]), and sporulation was almost abolished in both strains at 22˚C (Fig 3B). However, whereas sporulation was abolished in WT at 29˚C and 32˚C, some sporulation by the Δ*acb1* strain at 29˚C was evident (Fig 3B). Taken together, we conclude that *ACB1* is required for thermal adaptation to suboptimal and optimal but not elevated growth temperatures.

Further evidence that *ACB1* conditions responses to ambient temperature came when we assessed whether, like in Acb1-depleted yeast cells [19,20], the loss of *M. oryzae ACB1* affected membrane vesicular trafficking and membrane assembly. Endocytosis and intracellular membrane trafficking was monitored in WT and Δ*acb1* vegetative hyphae using the lipophilic selective dye FM4-64, which is affected in delivery to the vacuole compartment in yeast Acb1-depleted cells. Following growth in liquid MM for 20 h at the indicated temperatures, vegetative mycelia were exposed to FM4-64 for 2 mins [16]. FM4-64 was internalized in both WT and Δ*acb1* hyphae to stain endomembrane compartments (Fig 3C) indicating that, like in Acb1-depleted yeast strains [20], endocytosis was not affected by the loss of *ACB1* in *M. oryzae* (although we cannot entirely rule out that other FM4-64 uptake pathways such as those mediated by membrane-bound flippases are affected). FM4-64 uniformly stained WT and Δ*acb1 ACB1* cytoplasm at 22˚C– 29˚C and large vacuoles were evident at 26˚C and 29˚C. In contrast, Δ*acb1* vegetative hyphae accumulated FM4-64 in small puncta at 22˚C and 26˚C, and few vacuoles were visible at these temperatures (Fig 3C), which is consistent with observations of FM4-64 staining in yeast cells showing the accumulation of cytosolic vesicles [20,22] and vacuole morphology changes [20] in Acb1-depleted strains. However, at 29˚C, FM4-64 staining of Δ*acb1* resembled that of WT and the Δ*acb1 ACB1* complementation strain at 26˚C, with

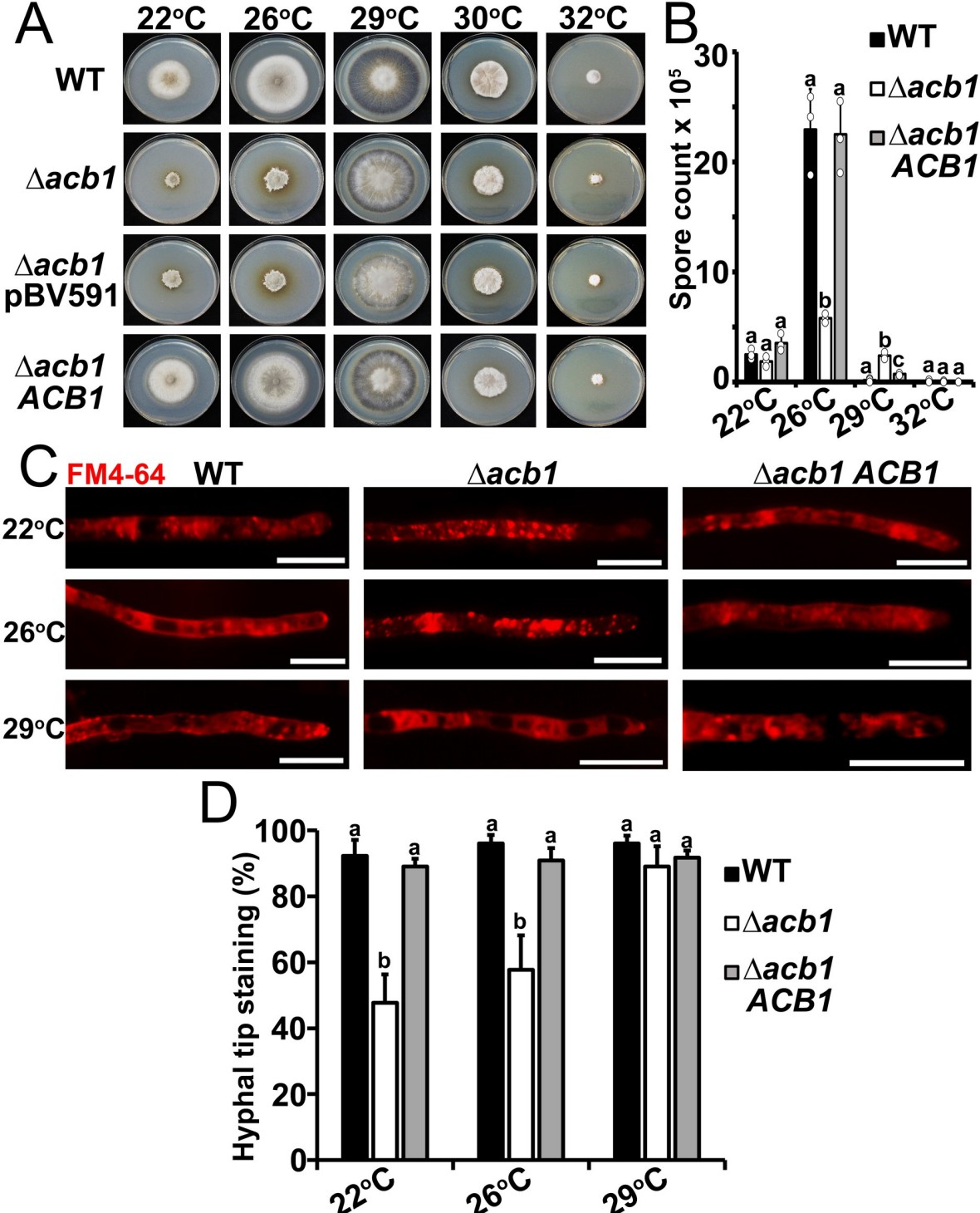

**Fig 3. *ACB1* disruption leads to temperature-dependent changes in growth, sporulation and membrane trafficking. A**. Radial growth of the indicated strains on complete media at the given temperatures after 10 days post inoculation (dpi). Images are representative of three biological replications. Δ*acb1* pBV591 is the mutant strain carrying the pBV591 vector to express *PWL2-mCherry:NLS* and *BAS4-GFP*. **B**. Mean counts ± SD (n = 3) of spores harvested at 10 dpi from CM at the indicated temperatures. Significant differences are denoted by lowercase letters (one-way ANOVA followed by Tukey HSD, *P* < 0.05). **C**. Micrographs of endocytosis by vegetative hyphae of the indicated stains following exposure to the amphiphilic styryl dye FM4-64. Cultures were grown for 24 h in CM then switched to minimal media for 20 h at the indicated temperatures, washed and stained with 1.3 mg/ml FM4-64 for 2 min at room temperature. Images are representative of three independent biological replicates. Scale bars, 10μm. **D**. Rates of hyphal tip staining by FM4-64. Cultures were grown in liquid CM at

26˚C for 24 h and then switched to liquid MM at the indicated temperatures for 20 h. Washed cells were stained with 1.3 mg/ml FM4-64 for 2 min at room temperature. Bars show means with error bars representing standard deviation. Different lowercase letters indicate significant differences (one-way ANOVA followed by Tukey HSD, $P < 0.05$). Values were obtained from observing 50 hyphal tips in each of three biological replicates at each temperature.

uniform FM4-64 cytoplasmic distribution, no puncta, and large vacuoles. By contrast, WT at 29˚C showed large vacuoles but some puncta, suggesting it was becoming somewhat stressed (Fig 3C). Thus, *ACB1* has a previously unreported temperature-dependent role in membrane trafficking and vacuole morphology. Furthermore, FM4-64 staining at hyphal tips was often depleted in Δ*acb1* vegetative hyphae at 22˚C and 26˚C but not at 29˚C, compared to WT and the Δ*acb1 ACB1* complementation strain (Fig 3C; quantified in Fig 3D), indicating changes in FM4-64 distribution in Δ*acb1* hyphae. This is consistent with defects in membrane trafficking in this strain at non-permissive temperatures and may also indicate a role for Acb1 in delivering vacuoles to sites of polarized growth. Together, consistent with yeast [20], vacuole morphological and membrane trafficking differences are observed between WT and Δ*acb1* at optimal and low growth temperatures, but, we first report here, at elevated temperatures, Δ*acb1* FM4-64 staining more closely resembles that of WT, suggesting *ACB1* is dispensable for membrane trafficking at elevated temperatures.

### *ACB1* is essential for rice blast disease and biotrophic growth at $\leq$ 26˚C but not at 29˚C

Considering vegetative growth and membrane dynamics were fully remediated in Δ*acb1* at 29˚C compared to WT, we hypothesized that the mutant strain would be restored for pathogenicity at elevated temperatures. However, Fig 4A shows that, when spores were applied to 3-weeks-old rice seedlings of the susceptible cultivar CO-39, virulence was almost abolished for all strains at 29˚C and 32˚C. At 22˚C, Δ*acb1* pathogenicity, but not WT or Δ*acb1 ACB1* pathogenicity, was abolished (although disease severity caused by WT and the complementation strain was reduced at 22˚C compared to 26˚C; disease lesions quantified in Fig 4B). Thus, although Δ*acb1* grew like WT at 29˚C, infection was not remediated at this temperature because both strains were reduced for virulence. However, using detached rice leaf sheaths, we determined that this was likely due to impaired appressorial penetration rates for all strains at 29˚C compared to 26˚C (Fig 4C). Although appressorial penetration rates were reduced in Δ*acb1* compared to WT at 26˚C (around 85% of WT appressoria had penetrated successfully by 36 hpi (n = 3) compared to about 75% for Δ*acb1*, Fig 4C *left*, which is in line with the observed reductions in turgor pressure in Δ*acb1* appressoria at this temperature (S1A Fig)), appressorial penetration rates were greatly reduced for both WT and Δ*acb1* at 29˚C (approx. 20% for WT vs approx. 15% for Δ*acb1*, Fig 4C *left*), which may also result from reduced appressorial turgor by all strains tested at this temperature compared to WT at 26˚C (S1B Fig). At 22˚C, WT appressorial penetration rates were reduced to around 60% while Δ*acb1* appressorial penetration rates, by 36 hpi, were around 15% (Fig 4C *left*). Therefore, considering 26˚C is the optimal temperature for both WT and Δ*acb1* appressorial penetration, we repeated the assay on detached rice leaf sheaths by inoculating all strains at 26˚C for 24 h (by which time mature appressoria have formed and generated penetration pegs [12]) before shifting the samples to 22˚C or 29˚C (or remaining at 26˚C) for an additional 12 h. Fig 4C *right* shows that following the temperature shift from 26˚C at 24 h, by 36 hpi, penetration rates for WT and Δ*acb1* at 29˚C were now not significantly different ($P > 0.05$), with both strains having penetration rates of approx. 80% and thus comparable to WT penetration under constant 26˚C temperatures. At 22˚C (after a temperature switch from 26˚C), WT and Δ*acb1* penetration rates were

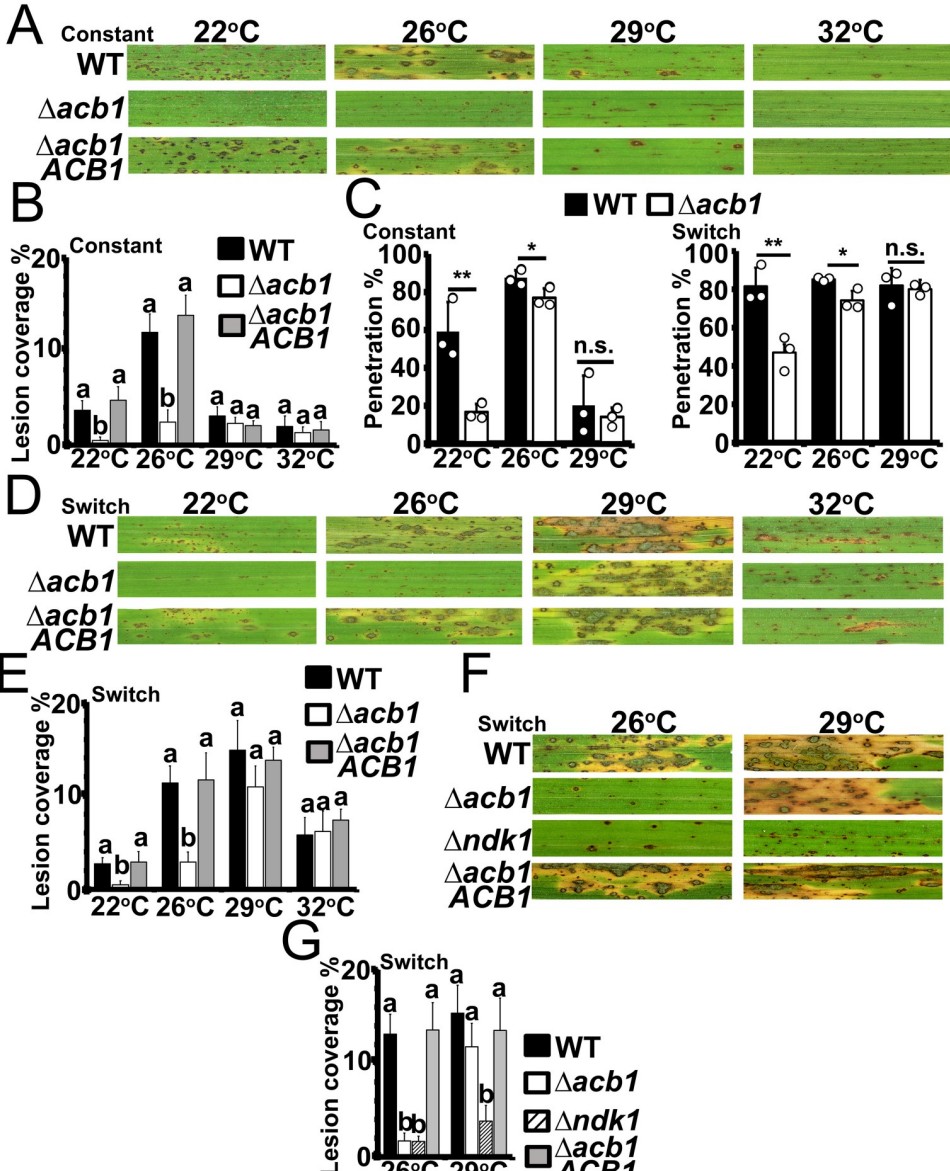

**Fig 4. *ACB1* is critical for rice blast disease at 22˚C and 26˚C but not at 29˚C. A.** Symptoms of rice blast disease on leaves of 3-weeks-old CO-39 rice seedlings after spraying with spores of the indicated strains at a rate of 1×10⁵ spores ml⁻¹. Inoculated plants were incubated for 120 hpi at the designated temperatures (ie. under constant temperature conditions). **B.** Lesion coverage rates on leaves of 3-week old rice seedlings infected with the indicated strains at 120 hpi at the indicated temperatures. Bars depict means ± SD. Different lowercase letters indicate significant differences (one-way ANOVA followed by Tukey HSD, $P < 0.05$). The experiment was performed twice, and calculations were determined from three biological replicates with three technical replicates each. **C.** Appressorial penetration rates for the indicated strains after incubation of inoculated detached rice leaf sheaths under constant temperatures for 36 hpi (*left*), or (*right*) after 24 h incubation at 26˚C followed by a temperature switch to the indicated temperatures for an additional 12 h. Spores of the indicated strains were applied at a rate of 3×10⁴ spores ml⁻¹. Bars depict means ± SD and were calculated from observing 50 infected rice cells per treatment, repeated in triplicate. Significant differences were determined by Student's t-test as indicated with asterisks: *$P < 0.05$, **$P < 0.01$; n.s. = not significant. **D.** Symptoms of rice blast disease on leaves of 3-weeks-old CO-39 rice seedlings after spraying with spores of the indicated strains at a rate of 1×10⁵ spores ml⁻¹. Plants were kept at 26˚C for 24 h before switching to the indicated temperatures for an additional 96 h. **E.** Lesion coverage rates on leaves of 3-week-old rice seedlings infected with the indicated strains. All samples were left at 26˚C for 24 h following spore inoculation before being switched to the indicated temperatures for 96 h. Bars depict means ± SD. Different lowercase letters indicate significant differences (one-way ANOVA followed by Tukey HSD, $P < 0.05$). Values are from three biological replicates with three technical replicates each. **F.** Symptoms of rice blast disease on leaves of 3-weeks-old CO-39 rice seedlings after spraying with spores of the indicated strains at

a rate of $1 \times 10^5$ spores ml$^{-1}$. Plants were kept at 26°C for 24 h before switching to the indicated temperatures for an additional 96 h. **G**. Lesion coverage rates on leaves of 3-week-old rice seedlings infected with the indicated strains. All samples were left at 26°C for 24 h following spore inoculation before being switched to the indicated temperatures for 96 h. Bars depict means ± SD. Different lowercase letters indicate significant differences (one-way ANOVA followed by Tukey HSD, $P < 0.05$). Values are from three biological replicates with three technical replicates each.

both increased compared to penetration under constant 22°C temperatures—to approx. 80% and 45%, respectively (Fig 4C **right**). Thus, switching from 26°C was best for WT and Δ*acb1* appressorial penetration. Next, we performed the temperature switch experiment on whole plants, whereby 3-week-old CO-39 rice seedlings were inoculated with spores and incubated at 26°C for 24 h before shifting the plants to 22°C or 29°C (or remaining at 26°C) for an additional 96 h. Fig 4D shows how, following the temperature shift from 26°C, Δ*acb1* pathogenicity was not restored at 22°C, but all strains became fully pathogenic at 29°C (quantified in Fig 4E).

We sought to ensure that improved infection at 29°C following a shift from 26°C for WT, Δ*acb1* and the Δ*acb1 ACB1* complementation strain was due to improved fungal colonization and not fungal-independent processes, such as reduced plant innate immunity responses following the temperature switch to 29°C compared to infection under constant 29°C temperatures. To achieve this, we repeated the temperature switch experiment from 26°C to 29°C (or constant 26°C) on whole leaves but this time incorporated the Δ*ndk1* mutant strain (Fig 4F; infection quantified in Fig 4G). The Δ*ndk1* mutant strain [26] is impaired in redox homeostasis and, although able to penetrate host cuticles at 26°C, is restricted for biotrophic growth in the first infected rice cell because it is unable to suppress the host oxidative burst, triggering host innate immunity. Suppressing the host oxidative burst by chemical means prevented host innate immune responses and remediated Δ*ndk1* biotrophic growth [26]. We reasoned that if better colonization of rice leaves by WT and Δ*acb1* at 29°C following the temperature shift from 26°C was due to impaired plant immune responses, then this should be reflected by better colonization by the Δ*ndk1* mutant strain at 29°C following the temperature switch compared to Δ*ndk1* infection under constant 26°C temperatures. However, Fig 4F shows that rice leaf infection by Δ*ndk1* (quantified in Fig 4G) was not remediated when switched to 29°C following penetration at 26°C, indicating that host infection by WT and Δ*acb1* at 29°C in temperature switch experiments is not due to impaired plant defense responses and that the remediation of Δ*acb1* at 29°C following the temperature shift from 26°C is specific to this mutant strain.

Together, we conclude that 26°C is optimal for penetration by both WT and Δ*acb1* appressoria, but, we report here for the first time, infection by both strains is devastating at 29°C when following a temperature shift from 26°C.

## BIC integrity is remediated in Δ*acb1* at 29°C

We next performed live-imaging cell imaging at 36 hpi on detached rice leaf sheaths that had been infected with WT and Δ*acb1* spores at 26°C for 24 h before switching the samples to 22°C or 29°C (or remaining at 26°C) for an additional 12 h (Fig 5A). Like under the constant 26°C control temperature, Δ*acb1* was reduced in biotrophic growth compared to WT at 22°C by 36 hpi, and demonstrated abnormal BIC morphologies (Fig 5A). However, at 29°C (following the temperature switch from 26°C), Δ*acb1* produced single WT-like punctate BICs in the first infected rice cell (Fig 5A) and BIC abnormality rates (a composite score of dual- and multi-BICs vs single BICs), were not significantly different in Δ*acb1* compared to WT by 36 hpi (Fig 5B; P > 0.05), where about 85% of IH displayed one BIC. Moreover, Bas4 intracellular retention in IH was not observed in Δ*acb1* at 29°C by 36 hpi after switching from 26°C (Fig

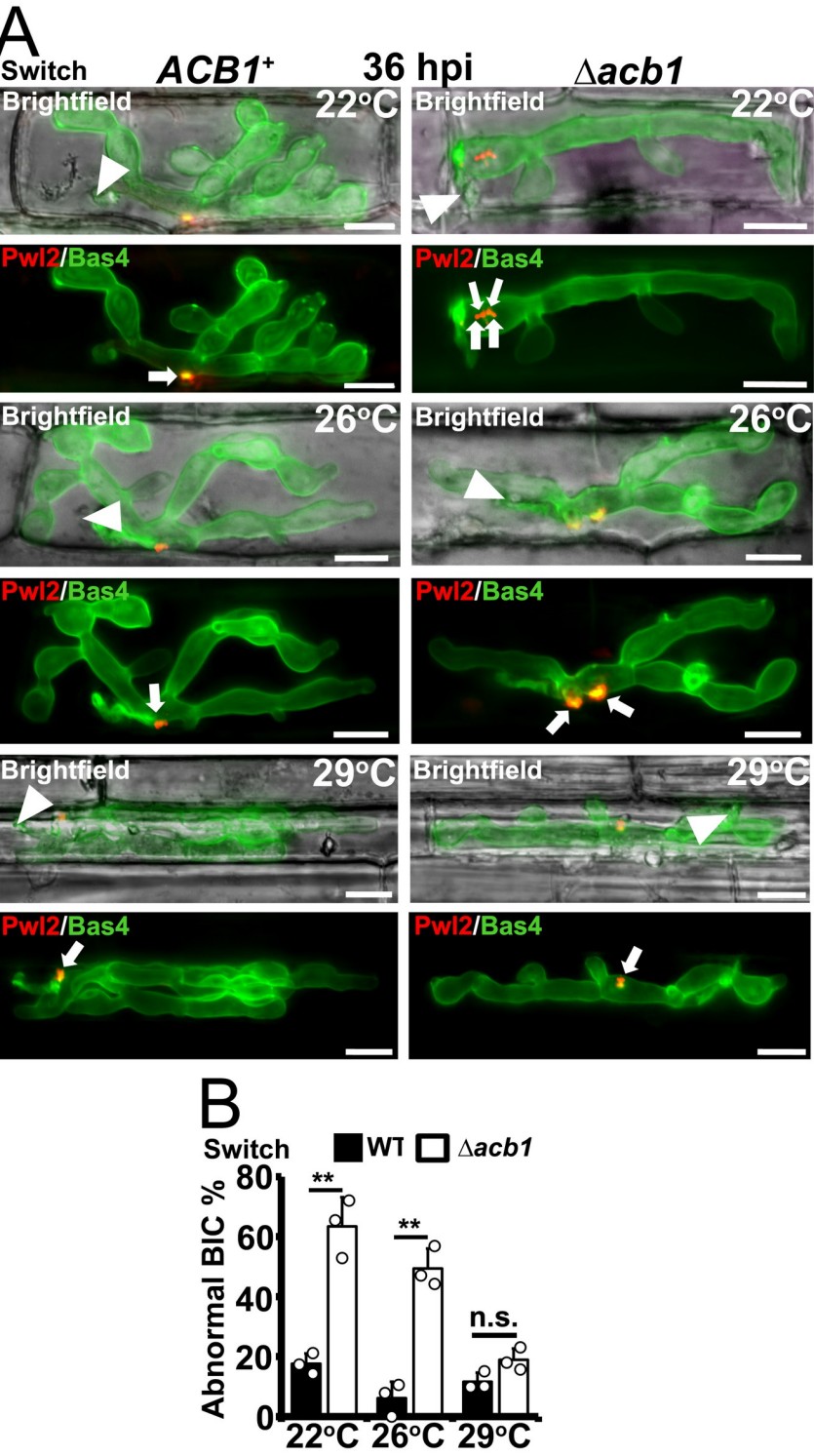

**Fig 5.** ***ACB1* is critical for BIC integrity at 22˚C and 26˚C but not at 29˚C. A**. Live-cell imaging of detached rice leaf sheaths inoculated at a rate of $3\times10^4$ spores ml[-1] with the indicated strains expressing *PWL2-mCherry:NLS* and *BAS4-GFP* and incubated under temperature-shift conditions of 24 h at 26˚C followed by a switch to the designated temperatures for an additional 12 h. Appressorial penetration sites are indicated with white arrowheads and white arrows show BIC localization; scale bars, 10μm. Images are representative of 50 infected rice cells per leaf sheath per strain, repeated in triplicate. **B**. Rate of occurrence of abnormal BICs in the indicated strains, including dual- and multi-BIC phenotypes, after the incubation of inoculated detached rice leaf sheaths at 26˚C for 24 h followed by a

temperature shift to the designated temperatures for an additional 12 h. Bars depict means ± SD and were calculated from observing 50 infected rice cells per treatment, repeated in triplicate. Significant differences were determined by Student's t-test as indicated with asterisks: *$P < 0.05$, **$P < 0.01$; n.s. = not significant.

5A). We also examined infected detached rice leaf sheaths at 44 hpi, following a temperature shift from 26˚C at 24 hpi (S3A and S3B Fig). Consistent with whole plant infection assays, by 44 hpi, infection at 29˚C (following the switch from 26˚C) resulted in Δ*acb1* IH filling the first infected cell and the remediation of tip-BIC morphology (S3A Fig), while the number of individual Δ*acb1* IH spreading cell-to-cell was increased compared to 26˚C and comparable to those observed for WT (S3B Fig). We conclude that, once penetration has proceeded successfully at 26˚C, Δ*acb1* is dispensable for biotrophic growth and rice blast disease at 29˚C. Furthermore, with regards to WT, our temperature shift experiments are the first to show how varying the infection temperature can affect WT disease severity, permitting *M. oryzae* to infect over a wider temperature range than previously appreciated, once the conditions of 26˚C for the first 24 hpi of penetration are met. This new information on host plant infection under temperature shift conditions could help improve disease prediction models.

## Membrane fluidity and membrane integrity is dependent on *ACB1* at optimal and suboptimal but not supraoptimal growth temperatures

We reasoned that increased growth and infection of Δ*acb1* at 29˚C compared to 26˚C resulted from remediation of lipid homeostasis at the elevated temperature. However, comparing the fatty acid profile of Δ*acb1* vegetative mycelia to WT at 22˚C, 26˚C and 29˚C revealed that, although there was a small but significant ($P < 0.05$) increase in total cellular fatty acids in Δ*acb1* at 32˚C, the abundances of total cellular fatty acids in Δ*acb1* was still reduced compared to WT at all other tested temperatures (S4A Fig). Furthermore, relative fatty acid abundance differences between Δ*acb1*, WT and Δ*acb1 ACB1* remained similar at 29˚C compared to 26˚C (S4B Fig). Thus, *ACB1* is required for lipid homeostasis at all tested temperatures, although perturbations are tolerated at 29˚C and 32˚C. Consistently, *ACB1* gene expression was constitutive under all temperatures tested (S5 Fig). We conclude that because lipid homeostasis was perturbed in Δ*acb1* compared to WT at all tested temperatures, the observed changes in fatty acid amounts and compositions in Δ*acb1* relative to WT can be tolerated at 29˚C but not at lower temperatures. Furthermore, the increase in total fatty acid amounts in Δ*acb1* at 32˚C compared to Δ*acb1* grown at the other temperatures may indicate that changes in fatty acid amounts in Δ*acb1* rather than changes in individual fatty acids are important for the Δ*acb1* phenotype.

We examined whether overexpressing *ACB1* affected host infection by complementing Δ*acb1* with a vector expressing *ACB1* under the constitutive *RP27* promoter. However, although there was a small but significant ($P < 0.05$) increase in total fatty acid amounts at 29˚C in Δ*acb1 pRP27::ACB1* compared to WT (S6A Fig), and some small but significant ($P < 0.05$) changes to the amounts of $C_{18:1}$ and $C_{18:3}$ at 26˚C (and other small changes at 22˚C, 29˚C and 32˚C) (S6B Fig), these changes had no effect on radial growth compared to WT or Δ*acb1 ACB1* at 22˚C, 29˚C and 32˚C (S7A Fig) or on host infection at 26˚C (S7B Fig), indicating that overexpressing *ACB1* has no discernable effect on fungal physiology under the conditions tested.

To account for the disparate features of the Δ*acb1* phenotype—such as the loss of thermal adaptation, impaired axenic and biotrophic growth, membrane trafficking defects and the loss of BIC integrity at 26˚C—and their full remediation at 29˚C, we postulated a role for *ACB1* in

homeoviscous adaptation at low temperatures. Homeoviscous adaptation is the organism's ability to maintain membrane fluidity within a narrow range independently of temperature [27,28]. Such a role for *ACB1* has not been previously described. However, because lipid homeostasis and fatty acid desaturation were still perturbed in Δ*acb1* at 29°C compared to WT, we considered that the ability of Δ*acb1* to maintain homeoviscous adaptation was impaired. Unsaturated fatty acid levels have a strong influence on membrane fluidity, but it is also influenced by many additional factors—including the relative abundances of polar head groups, sterols, sphingolipids and membrane proteins [27,29]. Thus, testing our hypothesis that *ACB1* was involved in homeoviscous adaptation necessitated measuring membrane fluidity directly. This was achieved by tracking temperature-dependent changes in lipid packing by monitoring the mobility of a membrane-bound fluorophore 1,6-diphenyl-1,3,5-hexatriene (DPH) [30] (Fig 6A). Membrane fluidity is inversely proportional to DPH fluorescence anisotropy such that higher DPH fluorescence anisotropy indicates lower membrane fluidity. To ensure that DPH fluorescence anisotropy corresponded directly with membrane fluidity in *M. oryzae* hyphae and not to some other unrelated variable(s), we first ran a positive control assay whereby mycelia were grown at constant 26°C, exposed to DPH for 40 min, and then treated with different concentrations of DMSO, which destabilizes and permeabilizes the cell membrane leading to increased membrane fluidity [31]. S8 Fig shows how as membrane fluidity increased due to increased DMSO exposure, DPH fluorescence decreased, thus confirming that DPH fluorescence is inversely proportional to membrane fluidity. Next, vegetative hyphae were grown in liquid CM at 26°C for 24h and then transferred to liquid MM at the indicated temperatures for 20 h before DPH was added. Fig 6A shows how, as expected, WT and the Δ*acb1 ACB1* complementation strain maintain membrane fluidity homeostasis across the 22°C—32°C temperature range, as indicated by similar anisotropy values. In contrast, Δ*acb1* anisotropy was significantly increased ($P < 0.05$) at 22°C and 26°C compared to the WT and the complementation mutant, indicating reduced membrane fluidity and an inability to maintain homeoviscous adaptation at these temperatures. However, at higher temperatures, Δ*acb1* anisotropy was not significantly different to the WT and/ or the complementation strain ($P > 0.05$), indicating that membrane fluidity in Δ*acb1* was remediated at elevated temperatures. Because anisotropy is a ratio reflecting DPH rotation (which does not occur in the cell wall), and DPH rotation is independent of DPH concentration in the membrane and thus unaffected by any potential differences in cell wall integrity between the strains, these results show that Δ*acb1* membranes are more rigid than WT at low temperatures but become more fluid as the temperature increases.

To corroborate our finding that membrane fluidity was significantly ($P < 0.05$) reduced in Δ*acb1* strains compared to WT, we next hypothesized that loss of membrane fluidity would affect plasma membrane integrity at 22°C and 26°C. To test this, we assessed membrane leakage of ions as an indicator of membrane integrity, expecting that compromised membranes in vegetative hyphae would show increased ion leakage. Fungal hyphae were grown for 40 h at 26°C before exposure to continuous increases in temperature in 2°C intervals from 22°C to 32°C. Electrolyte ion leakage was measured by recording the conductivity of cells in solution both after the incubation period and following boiling at 65°C to release all intracellular ions, allowing us to calculate fractional ion leakage at each temperature interval [32]. Between 22°C and 26°C, ion leakage was significantly ($P < 0.05$) increased in Δ*acb1* compared to the WT and the complementation mutant (Fig 6B), indicating that Δ*acb1* membranes leak more easily at low temperatures. However, at temperatures above 26°C, ion leakage from Δ*acb1* vegetative hyphae was comparable to the WT and the complementation strain ($P < 0.05$), suggesting that the observed defect in Δ*acb1* maintenance of homeoviscosity at lower temperatures results in less membrane integrity at those same temperatures.

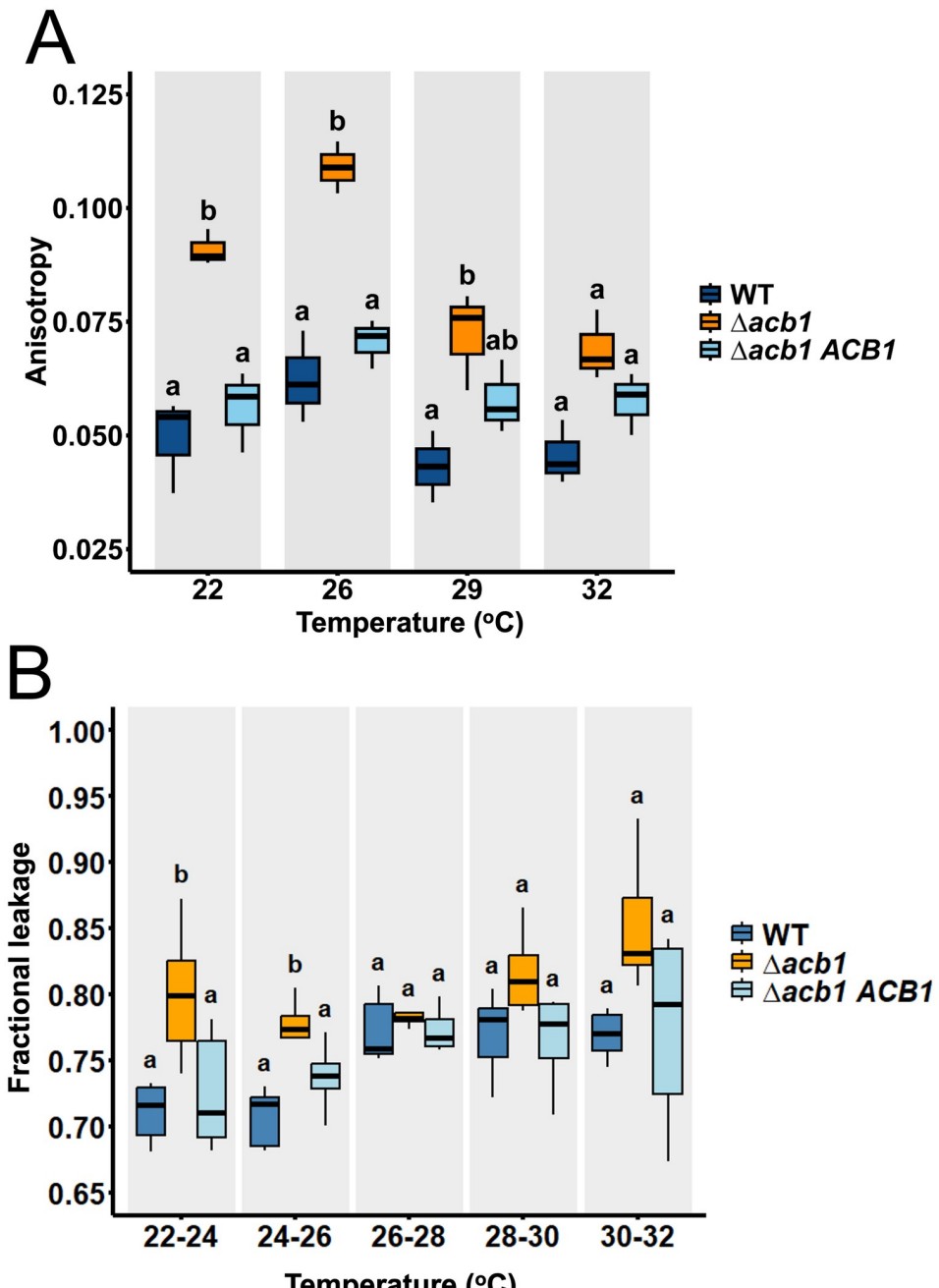

**Fig 6. Membrane fluidity and membrane integrity are dependent on *ACB1* at optimal and suboptimal temperatures. A**. Anisotropy was calculated from single-wavelength measurements of the fluorescent probe 1,6-diphenyl-1,3,5-hexatriene (DPH) using polarized excitation and emission filters and is inversely proportional to membrane fluidity. Cultures were grown in CM at 26°C for 24 h before switching to the indicated temperatures for an additional 20 h. After washing, hyphae were stained with 5 μM DPH for 40 min. Box plots display interquartile range (IQR), with boxes representing the 25th to 75th percentiles; the bold line inside the box is the mean value and whiskers extend to 1.5 times the IQR. Different lowercase letters indicate significance differences (one-way ANOVA followed by Tukey HSD, $P < 0.05$). The experiment was performed twice with each comprising three biological replicates with four technical replicates each. **B**. Fractional cellular electrolyte leakage measured in fungal hyphae of the indicated strains across a temperature gradient (22°C to 32°C). Fungal hyphae were grown at 26°C in liquid CM for 24 h and then transferred to liquid MM for 16 h. Thermal stress was simulated by gradually heating samples in a circulator. Samples were removed at 2°C intervals (from 22°C to 32°C) and analyzed for ion leakage, which is an indicator of membrane integrity. Box plots display interquartile range (IQR), with boxes representing the 25th to 75th percentiles; the bold line inside the box is the mean value and whiskers extend to 1.5 times the IQR. Different lowercase letters indicate

significant differences (one-way ANOVA followed by Tukey HSD, $P < 0.05$). The experiment was performed twice with each comprising three biological replicates with one technical replicate each.

We conclude that *ACB1*, via its role in maintaining lipid homeostasis, is required at optimal and lower temperatures for maintaining membrane homeoviscosity to prevent membrane rigidity and for maintaining plasma membrane integrity; however, the lipid changes arising from the loss of *ACB1* have the net effect of maintaining normal membrane fluidity, membrane integrity and membrane trafficking at supraoptimal temperatures relative to WT. Thus, homeoviscous adaptation and the control of membrane fluidity at optimal and lower temperatures requires *ACB1*, which is dispensable at elevated temperatures. This temperature-dependent role for *ACB1* in membrane fluidity is commensurate with its temperature-dependent role in hyphal growth and host infection.

## Discussion

Rice blast disease caused by *M. oryzae* is a threat to global rice and wheat harvests. Under climate change, its range and occurrence may change as temperatures warm and become more variable. Understanding how temperature extremes and climate variations affect *M. oryzae* host infection, and determining the underlying molecular mechanisms facilitating responses to ambient temperature and temperature changes during infection, will be critical to a better understanding of this devastating disease, potentially laying the groundwork for novel interventions suitable to a warming world. In this study, which focused on understanding membrane homeostasis in *M. oryzae* by characterizing the *ACB1* gene encoding the single *M. oryzae* acyl-CoA binding protein, we shed light on both these questions, firstly by showing how, following penetration at 26°C, the fungus can successfully establish disease at 29°C (and cause some symptoms at 32°C), thus widening our understanding of the thermal range of infection, and secondly, by determining how *ACB1* prevents membrane rigidity and the loss of membrane integrity at ≤ optimal growth temperatures to maintain membrane vesicular trafficking and, *in planta*, facilitate biotrophic growth and BIC organization.

Acb1 specifically binds *de novo* synthesized long-chain acyl-CoAs and transports them to downstream consumers such as membrane-bound fatty acid desaturases and elongases [19,20]. Similar to observations in yeast, total cellular fatty acid levels were reduced overall in Δ*acb1* compared to WT, and we observed relatively less $C_{16:0}$ and $C_{18:1}$ in Δ*acb1* than WT but more $C_{18:0}$ and $C_{18:3}$, together indicating that elongases and desaturases are active in the mutant strain, but their interaction with acyl-CoAs is perturbed in the absence of Acb1 transport. This may suggest that Acb1 regulates fatty acid interactions with elongases and desaturases. Although the precise mechanism of interaction between Acb1 and downstream enzymes is unknown in any system, developing interventions against *M. oryzae* elongases and desaturases would likely also affect the fatty acid pool and could be attractive targets in addition to Acb1 for abolishing infection.

*ACB1* was required for rice cell colonization at optimal and suboptimal infection temperatures but not at elevated temperatures, and the Δ*acb1* mutant strains grew axenically like WT at 29°C and higher—but not lower—temperatures. Membrane fluidity and membrane trafficking also did not require *ACB1* at 29°C and above, but homeoviscous adaptation [27] to prevent membrane rigidity at ≤ 26°C was *ACB1*-dependent. Because membrane fluidity and membrane trafficking but not fatty acid metabolism was remediated in Δ*acb1* at 29°C, our results suggest membrane fluidity is the key process, downstream of Acb1 (at 22°C -26°C) and upstream of membrane trafficking, underlying all the observed physiological changes in Δ*acb1*

strains compared to WT. At ≥ 29˚C, the fatty acid changes resulting from the loss of Δ*acb1* are tolerated with regards to membrane fluidity.

In WT, although membrane fluidity remained constant at the different temperatures studied, we showed for the first time that there was little change in the levels of measured unsaturated fatty acids as *M. oryzae* hyphae grew at elevated compared to optimal temperatures. This was unexpected as the double bonds in the hydrocarbon chain of unsaturated fatty acids are thought to cause kinks or bends that prevent the fatty acid chains from packing closely together, which increases the fluidity of the membrane. However, membrane fluidity is a complex phenomenon, poorly understood in fungi, that is influenced by many factors other than double bonds in fatty acids, such as the charge and size of head groups, chain lengths and non-glycerol lipid components (eg. sterols and proteins) [29,33]. Understanding these changes in *M. oryzae* membranes, which may underly the remediation of Δ*acb1* membrane fluidity at higher temperatures, is beyond the scope of the current paper, but warrants investigation in the future.

Our work is the first to report a role for *ACB1* in homeoviscous adaptation and previously, no determinants of *M. oryzae* thermal adaptation to optimal growth and infection temperatures were known. Although an *M. oryzae* mutant strain lacking *SLF1*—encoding a downstream transcription factor target of the MAP kinase Pmk1—was sensitive to high temperatures during axenic growth [34], our results are the first to characterize in any pathosystem a mutant fungal strain intolerant to normal and low but not high temperatures under both axenic and biotrophic growth conditions. This somewhat counter-intuitive insight into thermal adaptability and membrane fluidity enhances our knowledge of fungal infection processes and may be leveraged to open novel pathways to disease mitigation.

Optimal infection temperatures resulted in increased *ACB1*-independent appressorial penetration rates compared to penetration at suboptimal or supraoptimal temperatures. *ACB1* therefore ensures that the thermal range for invasive growth matches the optimum temperature for appressorial penetration. Once penetration is complete at 26˚C, *M. oryzae* causes disease on plant leaves at temperatures of up to 32˚C in a manner independent of *ACB1*. Thus, the infection process is more dynamic in response to temperature fluctuations than previously appreciated. Consequently, it may be necessary to account for day-to-day temperature fluctuations in fields exposed to *M. oryzae* when considering how rice blast disease might progress, and lab-determined disease-causing conditions for other phytopathogens should be reassessed by incorporating temperature changes after infection. For *M. oryzae*, integrating this new thermal range into disease prediction models may improve their accuracy. Also, understanding precisely why WT appressoria fail to penetrate efficiently at ≥ 29˚C, and whether they could mutate to penetrate at higher temperatures, is warranted. Recently, fluorescent probes were developed to measure membrane tension changes (the mechanosensor $N^+$-BDP) and changes in membrane chemical composition and lipid phase (the plasma membrane sensor NR12S) during appressorium formation [35]. Future work should deploy these probes for a comprehensive examination of the changes in plasma membrane rigidity and composition during appressorial development under different temperature regimes.

The loss of *ACB1* resulted in abnormal BIC formation at 26˚C but not 29˚C. Our results suggest that BIC organization relies on efficient membrane vesicle trafficking, which is itself dependent on membrane fluidity. Cytoplasmic effectors like Pwl2 are secreted into the BIC via an unconventional protein secretion (UPS) pathway [14, 15]. Although comprised of plant-derived membranes, the fusion of fungal vesicles (conceivably exosomes containing cytoplasmic effectors) to the BIC must occur across the fungal cell wall, and changes in the fluidity of fungal vesicle membranes might be expected to affect this process, subsequently disrupting BIC organization and vesicle interactions with host membranes. Furthermore, little is known about the

components of the *M. oryzae* UPS pathway, but a t-SNARE, Sso1, is required, and the loss of *SSO1* results in a similar dual-BIC phenotype to that observed here in Δ*acb1* strains [14]. t-SNAREs help fuse vesicles to the plasma membrane. In yeast Acb1-depleted cells, SNARE proteins are excluded from some intracellular compartments [20]. It is conceivable that similarly, due to the changes in membrane composition and fluidity, SNAREs may be absent or reduced in UPS vesicles (at ≤ 26˚C), leading to BIC perturbations. Furthermore, some SNAREs bind specific lipid headgroups and may be affected by changes to headgroup abundance in Δ*acb1* at ≤ 26˚C. However, how Sso1 regulates BIC organization, what other SNAREs may be involved, and what SNAREs decorate UPS vesicles are unknown. Also, it is possible that Acb1 may be directly involved, like Sso1, in the process of fusing UPS vesicles to the plasma membrane, for example by providing acyl-CoA to an acyl-CoA-dependent step during vesicle fusion. Such a direct role for Acb1 in vesicle fusion has been postulated in yeast with regards to the vacuole [22], but it has not yet (to our knowledge) been demonstrated. If occurring in *M. oryzae*, however, Acb1 would only be required for this direct fusion event at ≤ 26˚C.

In plants, membrane trafficking is important for the delivery of defense compounds to prevent pathogen invasion [36], but a role for plant *ACB1* homologues in this process is unknown. Moreover, in rice, decreased jasmonic acid signaling at 22˚C compared to 28˚C [37] compromised basal resistance, leading to more infection at the lower temperature than the higher temperature; this is similar to what we observe here for WT infection of whole plants at constant 22˚C vs constant 29˚C temperatures. Thus, there is a critical need to investigate how host plant membrane fluidity and/ or thermal adaptation may mediate immune responses to plant pathogens.

We conclude that acyl-CoA transport by the Acb1 protein prevents membrane rigidity and loss of membrane integrity at (sub)optimal temperatures, thereby maintaining membrane vesicle trafficking, but membrane fluidity does not require Acb1 at elevated temperatures. Our work provides a molecular-level account of how *M. oryzae* adapts to different temperatures to set the thermal range for host cell infection. This new knowledge has implications for our understanding and modelling of plant pathogen responses to global warming.

## Materials and methods

### Fungal strains and growth conditions

The *M. oryzae* Guy11 isolate served as the wild-type (WT) strain, and a full list of the strains used in this study can be found in S2 Table. Strains were maintained on complete media (CM) containing 1% (w/v) glucose, 0.2% (w/v) peptone, 0.1% (w/v) yeast extract and 0.1% (w/v) casamino acids [38], or on glucose minimal media (MM) [24]. After 10 days of incubation on plates at 22˚C, 26˚C, 29˚C, 30˚C or 32˚C with 12 h light-dark cycles, axenic growth was measured, and images taken with the Sony Cyber-shot digital camera (14.1 megapixels). For application, spores were harvested from 10-day-old CM plates and counted on a haemocytometer. For statistical analyses of sporulation rates, counts were performed on three biological replicates. For the cultivation of vegetative hyphae, unless otherwise stated, liquid cultures were grown at 150 rpm in CM for 40–44 hpi at the given temperature and washed with distilled water prior to any treatments.

### Functional gene analysis through targeted gene deletion and complementation

Disruption of the *ACB1* gene was achieved with the PCR-based split marker technique [24] using the primers listed in S3 Table. The coding region of *ACB1* was replaced with *ILV1*

conferring resistance to sulfonylurea. After transformation of fungal protoplasts [38], transformants were selected for sulfonylurea resistance and confirmed by PCR. At least three independent deletants were characterized fully and shown to be indistinguishable. One Δ*acb1* deletant was complemented with the full-length *ACB1* sequence under its native promoter to give the Δ*acb1 ACB1* complementation strain, or with the full-length *ACB1* sequence under the constitutive *RP27* promoter to give the Δ*acb1 pRP27::ACB1* complementation strain. The *ACB1*-carrying vectors were constructed with the *XhoI*-containing primers in S3 Table using pDL2 and the yeast GAP-repair method, as described by Zhou and colleagues [39].

## FAME analysis

Fatty acid methyl esters (FAMEs) were measured from lyophilized mycelia. Before harvesting, mycelia were grown for 24 h in liquid complete media followed by 16 h in liquid minimal media at 150 rpm under the indicated constant temperatures. Lyophilized mycelia were extracted and subjected to gas chromatography (GC), as previously described [40]. Briefly, 10–20 mg of lyophilized mycelia was transferred to a 1.8 ml GC autosampler vial containing 50 μl trimethyl sulfonium hydroxide (TMSH, 0.2 M in methanol) and 50 μg of pentadecanoic acid (15:0) as an internal standard. 500 μl of hexane (Sigma, USA) was added to the vial and capped. The vials were vigorously vortexed and incubated at room temperature for 15–30 minutes before being vortexed again. Samples were added to an Agilent autosampler tray (model 7693, Agilent, USA) and allowed to sit 10–15 minutes until the upper hexane phase containing the FAMEs separated from the debris. Samples were run on an Agilent 7890A GC system (Agilent, USA) containing a flame-ionization detector (FID). Fatty acids were identified by comparing retention times to standards. Samples were run in triplicate from two or three biological replicates.

## Incipient cytorrhysis assay

Appressorial turgor pressure were evaluated as previously described [12]. Spores from 10 days cultures were diluted to 1 x 10$^5$ spores ml$^{-1}$ and left to germinate on plastic coverslips for 24 hpi. Cell collapse rates for WT and Δ*acb1* strains were determined by counting under a microscope the number of appressoria undergoing cytorrhysis, characterized by the collapse of the appressorium, after treatment with increasing concentrations of glycerol (1–2 M) for 1 minute.

## Pathogenicity tests and live-cell imaging

For whole plant inoculations, spores of the indicated strains were harvested from 10-days-old CM plates, suspended in 0.2% gelatin (Acros, USA) and applied to 3-week-old rice seedlings of the susceptible cultivar CO-39 at a rate of $1 \times 10^5$ spores ml$^{-1}$. Inoculated plants were left at 22˚C for 3 h and then moved to a growth chamber with 12 h light-dark periods. For constant temperature treatments, plants were incubated at 22˚C, 26˚C, 29˚C or 32˚C for 120 h, while for temperature shift conditions, plants were kept at 26˚C for 24 h before being transferred to a growth chamber at 22˚C, 29˚C or 32˚C (or remaining at 26˚C) for 96 h. Recorded images were taken at 5 dpi. Lesion coverage rates at 120 hpi were quantified on whole infected rice leaves using ImageJ. Measurements were performed three times on three biological replicates. For live-cell imaging, detached rice leaf sheaths were inoculated with 3 x 10$^4$ spores ml$^{-1}$ per sample, in triplicate. For constant temperature treatments, leaf sheaths were incubated at 22˚C, 26˚C or 29˚C for 36 hpi or 44 hpi, as indicated, whereas in temperature shift treatments, inoculated samples were kept at 26˚C for 24h and then shifted to 22˚C or 29˚C (or remaining at 26˚C) for 12 hpi or 20 hpi, as indicated. Penetration rates were calculated from the number

of IH emerging from 50 appressoria per leaf sheath at 36 hpi, repeated in triplicate. The number of IH moving to neighboring cells was assessed at 44 hpi from 50 infected cells, repeated in triplicate. For DIC captures, leaf sheaths were imaged using a Nikon A1 confocal laser scanning microscope mounted on a Nikon 90i compound microscope at the University of Nebraska-Lincoln Morrison Microscopy Core Research Facility. For bright field and epifluorescence images, the Nikon Eclipse Ni-E upright microscope and NIS Elements software were used. Excitation/emission was 488 nm/505–531 nm for GFP and 543 nm/590–632 nm for mCherry.

## Visualizing endocytosis and membrane trafficking with FM4-64

Strains were grown in liquid MM at 22˚C, 26˚C or 29˚C at 150 rpm for 20 h. Vegetative hyphae was washed and stained with 1.3 mg/ml N-(3-Triethylammoniumpropyl)-4-(6-(4-(Diethylamino) Phenyl) Hexatrienyl) Pyridinium Dibromide (FM4-64; Invitrogen by Thermo Fisher Scientific) for 2 min at room temperature. Images were taken using the Nikon Eclipse Ni-E upright microscope and NIS Elements software. Excitation/emission was 543 nm/632 nm for FM4-64. Each treatment was repeated with three independent biological replicates. The rates of vegetative hyphal tip staining by FM4-64 were determined by counting the presence or absence of a fluorescent signal in at least 50 hyphal tips in each of three biological replicates at each temperature.

## Quantitative real-time PCR assay

For gene expression studies, WT vegetative hyphae were grown in liquid complete media at 26˚C for 24h, followed by a switch to minimal media at 22˚C, 26˚C or 29˚C for 16 h. Samples were lyophilyzed and RNA extracted using the RNeasy Plant Mini Kit (Qiagen, USA). DNAse treatment was done with DNAse I (Invitrogen, USA) and cDNA was generated using the qScript Supermix from QuantaBio (VWR International, USA). Quantitative real-time PCR was performed using the QuantiFast SYBR Green PCR Kit (Qiagen, USA), following Marroquin-Guzman and associates [41]. Fold changes were calculated using the $2^{-\Delta\Delta Ct}$ method [42].

## Fluorescence anisotropy measurements

Membrane fluidity was determined by capturing fluorescence anisotropy of mycelia stained with the fluorescent probe 1,6-diphenyl-1,3,5-hexatriene (DPH) (Acros Organics, UK). Hyphae were grown in liquid CM for at 26˚C and 150 rpm for 24 hpi and incubated at 22˚C, 26˚C, 29˚C or 32˚C for another 20 h. Samples were then washed, diluted to an $OD_{600}$ of 0.5–0.6 and stained with 5 μM DPH for 40 min at the designated temperature while shaking at 150 rpm. As a positive control, different concentrations of DMSO (0% -5% (v/v)) were added after DPH treatment to mycelia grown at 26˚C for 40 h; the samples were mixed and analyzed immediately. Single-wavelength measurements were performed at 355 nm (excitation) and 430 nm (emission) with light polarized in parallel and perpendicular orientation with respect to the polarization of the excitation light using a Cary Eclipse fluorescence spectrophotometer (Varian). Four measurements were obtained for each polarization and the sample was agitated in between to prevent the sedimentation of cells. Fluorescence anisotropy $r$ was calculated with $r = \frac{I_{\parallel} - I_{\perp}}{I_{\parallel} + 2I_{\perp}}$ [43] in which $I_{\parallel}$ and $I_{\perp}$ represent the observed intensities of light where the emission polarizer is oriented parallel ($\parallel$) or perpendicular ($\perp$) to the direction of the polarized excitation. To confirm that the observed signals reflected membrane fluidity, 0.5% DMSO was added post-measurement to promote a further drop in anisotropy. Background signal from unstained cells was subtracted from DPH-treated measurements for accuracy. This experiment

was repeated twice with each consisting of three biological replicates with four technical replicates each.

## Cellular electrolyte leakage assay

Cellular ion leakage by vegetative hyphae was determined by measuring conductivity under thermal stress conditions using a protocol modified from Shomo and colleagues [32]. Cultures were grown in liquid CM for 24 h at 26˚C and then shifted to liquid MM for an additional 16 h. Fungal hyphae were washed, split and transferred to 3 ml water. Afterward, samples were incubated in a refrigerated circulator (VWR, USA), starting at 22˚C and increasing at a rate of 1˚C / 15 min until 32˚C. At each 2˚C increment, a set of samples were removed and transferred to 50 ml tubes with 7 ml of water. Then, samples were equilibrated at room temperature for 20 min, followed by 20 min in a shaker at 230 rpm. Conductivity was measured with the Orion Star, A212, conductivity meter (Thermo Scientific, USA). Thereafter, samples were boiled by incubation in a water bath at 65˚C for 30 min. Samples were then cooled at room temperature for 40 min, with the final 20 min in a shaker at 230 rpm. A second measurement with the conductivity meter was taken to determine the maximum ion leakage. The fractional ion leakage for each temperature was calculated as the ratio between the initial and maximum conductivity readings. This experiment was repeated twice with each consisting of three biological replicates with one technical replicate each.

## Supporting information

**S1 Fig. Loss of *ACB1* results in reduced appressorial turgor.** Appressorial turgor was measured by the incipient cytorrhysis (cell collapse) assay using 1 M and 2 M glycerol after incubation of germinating spores at 26˚C (**A**) or 29˚C (**B**) for 24 h, by which time mature appressoria have formed. At least 100 appressoria from three biological replicates were observed to calculate the appressorial collapse rates. Error bars represent standard deviations and different lowercase letters are significantly different (unpaired t test with Bonferroni-correction, $P < 0.05$). (TIF)

**S2 Fig. Loss of *ACB1* reduces axenic growth on complete media and minimal media compared to WT at optimal and suboptimal but not supraoptimal temperatures. A**. Radial colony diameter values after growth on complete media plates at the indicated temperatures for 10 days. Bars depict means with error bars showing standard deviation. Different lowercase letters indicate significant differences (one-way ANOVA followed by Tukey HSD, $P < 0.05$). Values were obtained from three biological replicates and the experiment was repeated twice. **B**. Radial growth of the indicated strains on defined minimal media at the given temperatures after 10 days post inoculation. Images are representative of three biological replications. Δ*acb1* pBV591 is the mutant strain carrying the pBV591 vector expressing *PWL2-mCherry*:*NLS* and *BAS4-GFP*. (TIF)

**S3 Fig. Δ*acb1* IH growth and morphology is remediated at 29˚C by 44 hpi following a switch from 26˚C. A**. Live-cell imaging of detached rice leaf sheaths inoculated at a rate of $3 \times 10^4$ spores ml$^{-1}$ with the indicated strains expressing *PWL2-mCherry*:*NLS* and *BAS4-GFP* and incubated under temperature-shift conditions of 24 h at 26˚C followed by a switch to the designated temperatures for an additional or 20 h. Appressorial penetration sites are indicated with white arrowheads, white arrows show BIC localization and red arrows show Bas4 in intracellular compartments; scale bars, 10μm. Images are representative of 50 infected rice cells per leaf sheath per strain, repeated in triplicate. **B**. Mean numbers of individual hyphae of the

indicated strains moving from the first infected rice cell to neighboring cells by 44 hpi. Inoculated detached rice leaf sheaths were incubated at 26˚C for 24 h followed by a temperature shift to the designated temperatures for an additional 12 h. Bars depict means ± SD and were calculated from observing 50 infected rice cells per treatment, repeated in triplicate. Significant differences were determined by Student's t-test as indicated with asterisks: *$P < 0.05$, **$P < 0.01$; n.s. = not significant.
(TIF)

**S4 Fig. FAME analysis of mycelia from WT, Δ*acb1* and Δ*acb1 ACB1* strains grown at the indicated temperatures.** Total (**A**) and relative (**B**) amounts of the five most abundant fatty acids determined by FAME analysis following growth of vegetative mycelia incubated at 26˚C for 24 h in complete media followed by 16 h in minimal media at 22˚C, 26˚C, 29˚C or 32˚C. Values at 26˚C from Fig 1A are included for comparison. Bars depict the means of three biological replicates (raw data in S1 Table). Error bars are SD. Different lowercase letters indicate statistical differences as determined by one-way ANOVA followed by an unpaired t test with Bonferroni correction, $P < 0.05$. Asterisks (*) depict marginal significance, $0.05 < P < 0.06$. Black bars are WT, open bars are Δ*acb1* and grey bars are Δ*acb1 ACB1* strains.
(TIF)

**S5 Fig. *ACB1* expression is constitutive across temperatures.** To determine whether *ACB1* gene expression changed in response to temperature, quantitative real-time PCR (qPCR) was applied to cDNAs obtained from vegetative hyphae of the wildtype Guy11 strain grown for 24 h in liquid complete media at 26˚C followed by a switch to liquid minimal media at the indicated temperatures for an additional 16 h. *ACB1* expression levels were normalized against expression of the *M. oryzae* actin-encoding gene *ACT1*. Template controls showed no amplification. Error bars indicate standard deviation and white circles represent data points of biological replicates with four technical replicates. One-way ANOVA p-value is displayed above the bars. qPCR assays were repeated twice.
(TIF)

**S6 Fig. FAME analysis of mycelia from WT and *pRP27::ACB1* strains grown at the indicated temperatures.** Total (**A**) and relative (**B**) amounts of the five most abundant fatty acids determined by FAME analysis following growth of vegetative mycelia incubated at 26˚C for 24 h in complete media followed by 16 h in minimal media at 22˚C, 26˚C, 29˚C or 32˚C. WT values from S4 Fig are included for comparison. Bars depict the means of three biological replicates (raw data in S1 Table). Error bars are SD. Different lowercase letters indicate statistical differences as determined by one-way ANOVA followed by an unpaired t test with Bonferroni correction, $P < 0.05$. Asterisk (*) depicts marginal significance, $P < 0.06$. Black bars are WT and open bars are the *pRP27::ACB1* complementation strain.
(TIF)

**S7 Fig. Constitutive expression of *ACB1* does not affect radial growth or infection compared to WT. A**. Radial growth of the indicated strains on complete media after 11 days growth at the indicated temperatures. Images are representative of three biological replicates. **B**. Symptoms of disease at 26˚C after 120 hpi (*top*). 3-weeks old rice seedlings were sprayed with 1 x $10^5$ spores ml$^{-1}$ of the indicated strains, repeated three times. Relative lesion coverage was quantified (*bottom*) using infected leaves of 3-week old rice seedlings after 120 hpi at 26˚C. Bars depict means ± SD. Different lowercase letters indicate significant differences (one-way ANOVA followed by Tukey HSD, $P < 0.05$). Calculations were determined from three biological replicates with three technical replicates each.
(TIF)

**S8 Fig. Effect of DMSO on DPH fluorescence as a positive control for the membrane fluidity assay.** To ensure that single-wavelength measurements of the fluorescent probe 1,6-Diphenyl-1,3,5-hexatriene (DPH) using polarized excitation and emission filters, that were used for anisotropy calculations, corresponded with membrane fluidity, DMSO, which destabilizes and permeabilizes the cell membrane leading to increased membrane fluidity, was added in concentrations of 0% -5% (v/v) to WT mycelia that had been stained with DPH for 40 min. Cultures were grown at 26˚C for 40 h. Bars depict means ± SD. The experiment was performed on three biological replicates with four technical replications each.
(TIF)

**S1 Table. Fatty acid methyl ester (FAME) analysis of mycelia from the indicated strains.**
(XLSX)

**S2 Table. Strains used in this study.**
(DOCX)

**S3 Table. Oligonucleotides used in this study.**
(DOCX)

**S4 Table. Raw data behind the graphs.**
(XLSX)

## Acknowledgments

We thank Ngoc T. T. Pham and Jocelyne Horanituze, Department of Plant Pathology, UNL, Dr. Javier Seravalli, Redox Biology Center, UNL, and Fangye Li, Department of Biochemistry, UNL, for technical support.

## Author Contributions

**Conceptualization:** Richard A. Wilson.

**Data curation:** Michael Richter, Wayne R. Riekhof, Richard A. Wilson.

**Formal analysis:** Michael Richter, Richard A. Wilson.

**Funding acquisition:** Richard A. Wilson.

**Investigation:** Michael Richter, Lauren M. Segal, Raquel O. Rocha, Nisha Rokaya, Wayne R. Riekhof.

**Methodology:** Michael Richter, Aline R. de Queiroz, Wayne R. Riekhof, Rebecca L. Roston.

**Project administration:** Richard A. Wilson.

**Resources:** Wayne R. Riekhof, Rebecca L. Roston, Richard A. Wilson.

**Supervision:** Richard A. Wilson.

**Validation:** Michael Richter, Richard A. Wilson.

**Visualization:** Michael Richter, Richard A. Wilson.

**Writing – original draft:** Michael Richter, Richard A. Wilson.

**Writing – review & editing:** Michael Richter, Rebecca L. Roston, Richard A. Wilson.

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
