## [Decision Letter · Decision Letter 0]

1 Jul 2024

Dear Dr. Wilson,

dear Rich,

Thank you very much for submitting your manuscript "Membrane fluidity control by the Magnaporthe oryzae acyl-CoA binding protein sets the thermal range for host rice cell colonization" for consideration at PLOS Pathogens. As with all papers reviewed by the journal, your manuscript was reviewed by members of the editorial board and by several independent reviewers. In light of the reviews (below this email), we would like to invite the resubmission of a significantly-revised version that takes into account the reviewers' comments.

We cannot make any decision about publication until we have seen the revised manuscript and your response to the reviewers' comments. Your revised manuscript is also likely to be sent to reviewers for further evaluation.

Sincerely,

Bart P.H.J. Thomma

Section Editor

PLOS Pathogens

Bart Thomma

Section Editor

PLOS Pathogens

Michael Malim

Editor-in-Chief

PLOS Pathogens

orcid.org/0000-0002-7699-2064

Reviewer's Responses to Questions

**Part I - Summary**

Reviewer #1: In this study the authors Richter et al., have investigated what molecular and cellular processes are required to determine the temperature range for successful biotrophic growth of the rice blast fungus Magnaporthe oryzae. This study is considerably timely considering the imposing threat climate change is having on the environment, and how these changes could effect the spread of destructive plant pathogens globally. In order to understand how the fungus may adapt to changes in temperature, the authors characterised a gene encoding an acyl-CoA binding protein ACB1, an intracellular transporter of long chain acyl-CoA esters. The mutant displayed aberrant fatty acid desaturation levels compared to wild type, and loss of pathogenicity at low (22°C) and optimal (26°C) temperatures, but this phenotype was remediated at elevated (29°C). Further analysis revealed the mutant was reduced in vegetative growth and impaired membrane trafficking at 22°C and 26°C, but not at 29°C. During biotrophic growth the mutant displayed a multi-BIC phenotype and inhibited growth at 26°C, which was remediated at 29°C. Finally, the authors showed the ∆acb1 mutant had impaired membrane fluidity at 22°C and 26°C but not elevated temperatures, suggesting ACB1 suppresses membrane rigidity at optimal and suboptimal but not elevated temperatures. In summary, the authors suggest ACB1 helps to keep the cell membrane flexible and less rigid at 22°C and 26°C, which is important for cell function. However, at higher temperatures, ACB1 is dispensable for this function. I find this study very interesting as it is suggesting rice blast disease can survive in a wider range of temperatures than previously thought. However, I do have some points that should be addressed before publication can be considered.

Reviewer #2: This manuscript functional characterized the ACB1 gene encoding the single acyl-CoA-binding protein, an intracellular transporter of long-chain acyl-CoA esters, in Magnaporthe oryzae. The manuscript is very well-written and the data is presented in a clear and concise fashion. The introduction and discussion are rich in fundamental background information. The experimental designs, results, and conclusions are very consistent and the outcomes should serve as excellent references for fungal geneticists and plant pathologists. Thank you for putting together this fantastic research data that leads to a high-quality manuscript.

I strongly support the publication of this manuscript in the journal Plos Pathogens.

Reviewer #3: Strengths

This is a well-written, novel, and thought-provoking study that addresses the importance of an acyl-coA binding protein in influencing the thermal constraints of vegetative growth and biotrophic infection of rice by the blast fungus Magnaporthe oryzae. The authors describe the phenotypic characterization of a previously described acb1 knockout mutant with altered fatty acid membrane composition, that is perturbed in vegetative growth and virulence at normally optimal temperatures (22-26 degrees C). The authors demonstrate convincingly using live-cell imaging that acb1 KO mutants produce atypical BIC's, which are sites of cytoplasmic effector secretion, at these low, normally optimal temperatures. The authors also provide (less robust) data to support the idea that membrane trafficking is perturbed at lower, but not higher, temperatures in acb1 KO mutants. Interestingly, the authors go on to convincingly demonstrate that acb1 KO mutants can actually cause disease at high temperatures (but not low) when shifted to these higher temperatures after appressorium-mediated cuticle penetration. The authors go on to provide data from a membrane fluidity assay, to support the idea that the acb1 protein is important for maintaining membrane fluidity at lower and optimal, but not high temperatures. Overall, the authors conclude that acb1 has a key role in membrane remodeling (homoviscous adaptation) at lower and optimal temperatures, but is dispensable at elevated temperatures.

To my knowledge, this is the first study of its kind to consider thermal constraints on biotrophic colonization and infection by a fungal pathogen, which is a major strength. I think the manuscript has the potential to make a significant impact on the field, and get the plant-microbe community thinking about the nature of these interactions beyond optimal growth chamber conditions etc. However, I feel a couple of key experiments need further quantitative support.

Weaknesses

A weakness in the manuscript is a lack of quantitation for the FM4-64 membrane trafficking assay (Figure 2C) and the plant infection assays (Fig 2 and 3) - both of which should be relatively easy to address, I hope. I also think the anisotropy assay, given its key importance to the manuscript, needs further explanation in the methods, and possibly some further controls (as outlined in Part II). It would also be compelling to see the fatty acid abundance of a genetically complemented acb1 KO strain return to wild type- like composition, and some of the other important phenotypes (e.g. multiple BIC's) rescued by complementation.

**Part II – Major Issues: Key Experiments Required for Acceptance**

Reviewer #1: The FAME analysis performed in Figure 1A, B should also be performed on the complemented strain.

In Figure 3 the authors show that switching to 29°C restores pathogenicity in the mutant. It has been shown that increasing temperature of rice can disrupt the function of some NLR’s. Can this be discounted? Constant incubation at the higher temperatures leading to no infection for all strains could be due to stress of strains. Perhaps in the supplementary data the authors could include another non related mutant that has a similar reduction in pathogenicity at 26°C, and shift it to 29°C to check it’s pathogenicity isn’t restored? I would like to see that this phenomenon the authors report is specific to this mutant only.

What happens to the second acetylCoA gene expression at higher temperatures in the absence of ACB1, does it compensate in any way? The authors suggest it doesn’t because the lipid composition still remains perturbed at 29°C, however, with a perturbed lipid composition at 29°C the mutant can now cause disease. Is that because 29°C increases membrane fluidity in the mutant? For me this wasn’t clear, and I think generation of the double mutant would help clarify this point.

What happens to ACB1 expression at higher temperatures? Is still expressed and temperature overrides its requirement?

The authors claim that loss of ACB1 raised the level of unsaturated fatty acids compared to wild type. An increase in unsaturated fatty acids is generally associated with the membrane being more fluid, but the authors suggest loss of ACB1 leads to a lower membrane fluidity at lower temperatures compared to wild type (based on later anisotropy experiments)? Unsaturated fatty acids have one or more double bonds in their hydrocarbon chains, causing kinks or bends. These kinks prevent the fatty acid chains from packing closely together, which increases the fluidity of the membrane. Can the authors comment?

The authors show that appressoria can normally function, in that they can generate enough pressure to penetrate the host leaf cuticle. Does that mean the role of ACB1 in lipid homeostasis is specific to invasive hyphal growth/BIC formation? I would have thought the alterations in membrane composition would also influence appressorium function? Cao et al., 2023 also characterised this gene and suggested the mutant displayed a reduction in appressorium specific turgor pressure.

What happens if ACB1 is overexpressed? Is this strain more pathogenic at lower temperatures and compromised at higher temperatures?

Line 153 the authors say ‘In contrast, dacb1 IH accumulated…’ however it is vegetative hyphae displayed in Figure 2C? The authors suggest in line 151 that the loss of ACB1 did not effect endocytosis. However, have the authors considered what other proteins may be effected in function at low and optimal growth conditions in the mutant? For example, other membrane associated proteins like flippases? In fact, it has been reported that FM4-64 internalisation can be taken up via two pathways, by endocytosis or by internalisation involving flippase and lipid transfer protein activity, which can lead to different patterns of staining. Although endocytosis may remain unaffected, membrane bound proteins such as flippases may effected in the mutant?

The images in Figure 2C are not very convincing for me, and I feel the conclusion being drawn, that ACB1 has a temperature dependent role in membrane trafficking is a strong conclusion based on the images/data provided. Do GFP tagged membrane trafficking proteins, for instance, display a change in their localisation pattern in the mutant at 22 and 26 compared to WT to verify what is being observed with the FM4-64 stain? These images need quantifying. It also seems like z stack projections have either not been provided, or the tips of hyphae have not stained well, or in the mutant strain the tips of hyphae are aberrant- for example is the spitzenkorper disrupted?

The authors show by DPH fluorescence anisotropy that the WT at 22-32 maintains membrane fluidity homeostasis. However, FAME experiments show that the WT fatty acid composition changes as you incubate 22-29. Statistical analysis needs to be performed between wild type samples as temperature changes for the FAME experiment in S3. Are these two results contradicting? Altered fatty acid composition effects the levels of saturated vs unsaturated fatty acids within the membrane which in turn effects the membrane fluidity? Please could the authors comment?

In order to carry out the staining cells were permeabilised with Amphotericin B, can the authors to sure that the increased fluorescence observed in the mutant isn’t due to other effects knocking out the gene may have on the cell wall for example, which could lead to enhanced permeabilization and subsequent uptake of dye in the mutant vs WT?

Why do the authors not provide images of the samples that were used to generate the graph?

I would like to see all the data points and replicates on the graph. N=3 how did the authors take measurements, did they take an average of multiple technical reps and then repeat this 3 times? The error bars are also very large. For me this is the weakest part of the manuscript and this experiment needs revising.

Reviewer #2: (No Response)

Reviewer #3: Fig 2. I don't think the data presented in this figure is sufficient to support the conclusion that membrane trafficking is altered in acb1 mutants. I think the FM4-64 assay in principle is okay, but the authors need to find a way to make this quantitative and more robust, rather than showing single representative images. It would also be more compelling to include data to show that the complementation strain is restored to wildtype-like behavior in this assay.

I also think the authors should make some effort to quantify disease symptoms on leaves to increase the robustness of these findings. For example quantifying lesion size etc.

Fig 4. I would like to see the individual data points plotted for this graph, and a more complete description of the assay in the methods. It's not clear what a replicate is in this experiment, or precisely how the measurements were taken. Is this assay done in cuvettes using mycelia? Are the mycelium homogenized etc? I can see how this assay would work for single-celled yeast, but it's less obvious how it would work with filamentous fungi. I'd also like to see more biological replicates if possible - the error bars for the mutant, especially at the higher temperature are quite large. The conclusion that membrane fluidity in acb1 mutants at high temperatures is not significantly different from WT at 29 and 32 degrees, needs strengthening. Ideally, and perhaps this is asking too much, I'd like to see further evidence that this assay can reliably detect changes in membrane fluidity in control hyphae - perhaps in response to drugs known to perturb membrane properties, as in the cited studies.

I'm hoping that this will be relatively straightforward for the authors to do, and will make these important findings more robust.

**Part III – Minor Issues: Editorial and Data Presentation Modifications**

Reviewer #1: Figure 1D the multiple BICs highlighted in the red boxes need quantifying. I think it would be good to have a zoomed in micrograph of the wild type BIC phenotype for comparison for readers who are less familiar with this topic.

To complement the FAME analysis in Figure 1A, B have the authors considered to quantify/visualise changes in the plasma membrane ridgity and composition of vegetative hyphae and appressoria using the probes N+-BDP (for tension changes) and NR12S (chemical polarity and lipid phase) published by Michels et al., 2021?

Have the authors considered making a model summarising their proposed function of ACB1?

Reviewer #2: Figure 2A: Please enlarge the data points (circles) in the bar graph.

Figure 2A: The growth assay illustration is extremely convincing. The authors could also show a graph showing the three replicates of the growth assay.

Figure 3A and C: The virulence assay data is very convincing. However, if the authors could show the quantification, such as number of lesions/leaf area or diseased area, it would make the data even more convincing.

Reviewer #3: I'm a little curious as to why WT replicates were omitted from the FAME analysis at 26 and 29 degrees.

PLOS authors have the option to publish the peer review history of their article (what does this mean?). If published, this will include your full peer review and any attached files.

Reviewer #1: No

Reviewer #2: No

Reviewer #3: No
---

## [Decision Letter · Decision Letter 1]

10 Nov 2024

Dear Dr. Wilson,

We are pleased to inform you that your manuscript 'Membrane fluidity control by the Magnaporthe oryzae acyl-CoA binding protein sets the thermal range for host rice cell colonization' has been provisionally accepted for publication in PLOS Pathogens.

Best regards,

Bart P.H.J. Thomma

Section Editor

PLOS Pathogens

Michael Malim

Editor-in-Chief

PLOS Pathogens

orcid.org/0000-0002-7699-2064

Reviewer Comments (if any, and for reference):

Reviewer's Responses to Questions

**Part I - Summary**

Reviewer #1: I am happy with the amendments the authors have made to their manuscript, and believe it should now be accepted for publication. Nice work everyone!

Reviewer #3: I thank the authors for carefully addressing all of my concerns - the manuscript is much improved.

**Part II – Major Issues: Key Experiments Required for Acceptance**

Reviewer #1: N/A

Reviewer #3: None

**Part III – Minor Issues: Editorial and Data Presentation Modifications**

Reviewer #1: N/A

Reviewer #3: (No Response)

PLOS authors have the option to publish the peer review history of their article (what does this mean?). If published, this will include your full peer review and any attached files.

Reviewer #1: No

Reviewer #3: No

---

## [Editor Report · Acceptance letter]

18 Nov 2024

Dear Dr. Wilson,

We are delighted to inform you that your manuscript, "Membrane fluidity control by the Magnaporthe oryzae acyl-CoA binding protein sets the thermal range for host rice cell colonization," has been formally accepted for publication in PLOS Pathogens.

Best regards,

Michael Malim

Editor-in-Chief

PLOS Pathogens

orcid.org/0000-0002-7699-2064